# COVID-19 Pandemic: Analysis of Health Effects on Emergency Service Nursing Workers via a Qualitative Approach

**DOI:** 10.3390/ijerph20064675

**Published:** 2023-03-07

**Authors:** Elaine Cristine da Conceição Vianna, Raquel Veiga Baptista, Raquel Silva Gomes, Gabrielle Silva Pereira, Giovanna Costa Guimarães, Magda Guimarães de Araujo Faria, João Silvestre Silva-Junior, Marcelia Cristina de Oliveira, Luana Cardoso Pestana, Daniela Campos de Andrade Lourenção, Mirian Cristina dos Santos Almeida, Vivian Aline Mininel, Silmar Maria da Silva, Aline Coutinho Sento Sé, Cristiane Helena Gallasch

**Affiliations:** 1Nursing School, Universidade do Estado do Rio de Janeiro, Rio de Janeiro 20551-030, RJ, Brazil; 2Department of Public Health Nursing, Nursing School, Universidade do Estado do Rio de Janeiro, Rio de Janeiro 20551-030, RJ, Brazil; 3Department of Medicine, Centro Universitário São Camilo, São Paulo 04262-200, SP, Brazil; 4Department of Forensic Medicine, Bioethics, Occupational Medicine, Physical Medicine and Rehabilitation, Medical School, Universidade de São Paulo, São Paulo 01246-903, SP, Brazil; 5Hospital Federal Cardoso Fontes, Ministério da Saúde, Rio de Janeiro 22745-130, RJ, Brazil; 6Nursing School, Universidade de São Paulo, São Paulo 05403-000, SP, Brazil; 7Nursing Undergraduation Course, Universidade Federal do Tocantins, Palmas 77001-092, TO, Brazil; 8Department of Nursing, Centro de Ciências Biológicas e da Saúde, Universidade Federal de São Carlos, São Carlos 13565-905, SP, Brazil; 9Department of Basic Nursing, School of Nursing, Universidade Federal de Minas Gerais, Belo Horizonte 30130-100, MG, Brazil; 10Department of Medical-Surgical Nursing, Nursing School, Universidade do Estado do Rio de Janeiro, Rio de Janeiro 20551-030, RJ, Brazil

**Keywords:** workers’ health, occupational exposure, nursing workers, hospital emergency services, COVID-19, illness

## Abstract

During the COVID-19 pandemic, longstanding issues involving nursing work, which has always involved significant risks of illness and infection, were intensified. It is necessary to acknowledge the risks and nurses’ perceptions about the risks qualitatively during the period of the health crisis. The aim of this study was to examine the health repercussions perceived by nursing workers in emergency services during the outbreak of the COVID-19 pandemic in Brazil. This was a qualitative, descriptive, cross-sectional study. The settings of the study were emergency services with a national scope; the participants were nursing workers. Data were collected via face-to-face virtual calling interviews and analyzed via a content analysis technique, which was supported by IRAMUTEQ software. The formation of textual classes pointed in three thematic directions, from which three categories emerged: nursing workers’ exposure, due to a lack of protective equipment, to the risk of being contaminated with, falling ill from, and transmitting the COVID-19 virus; changes in work environments, processes, and relations in response to the pandemic; and physical, mental, and psychosocial alterations perceived by emergency service nursing workers. The exposure to the virus, risk of contamination, and changes in the work environment and relations all resulted in health repercussions, which were perceived as physical, mental, and psychosocial alterations that were described as dietary disturbances, physical fatigue, burnout, increased smoking, anxiety, sleep alterations, fear, exhaustion, stress, social isolation, loneliness, distancing from relatives, and social stigma.

## 1. Introduction

On 11 March 2020, the World Health Organization declared COVID-19 to be a pandemic, raising fears in workforces, especially among nursing workers, regarding the rapid spread and magnitude of the disease caused by the new coronavirus [1].

Researchers have pointed to the social and labor repercussions for health workers in that scenario, given the need to review protocols, alter flows, and introduce strategies to prevent the disease caused by the 2019 coronavirus among workers exposed during their labor activities [1,2].

Prominent developments from the pandemic included worsening health, economic, social, labor, and humanitarian crises, which strongly affected nurses, who experienced social inequalities heightened by exploitation of their labor and restrictions on their collective action as workers [3].

The literature shows that large numbers of health workers were infected by the coronavirus because healthcare-associated transmission was favored by inadequate protection. Among those workers, nursing was one of the health profession categories most affected by COVID-19 [4,5].

Illness among nursing workers is monitored by the Nursing Observatory, and figures released by the Federal Nursing Council indicate that 64,610 cases of illness had been recorded by 19 September 2022, with 872 deaths [6].

In Brazil, as in other countries, during the COVID-19 pandemic, emergency services were responsible for treating patients with symptoms suggestive of the disease, which placed those services under heavy demand for care [7].

Emergency services, an important component of the healthcare system, are governed in Brazil by the National Emergency Care Policy and the Technical Regulations of State Emergency Systems [8]. They are recognized not only for their importance to the public’s access to Brazil’s unified health system (SUS) but also for their essential inherent characteristics [9]. Emergency services form part of the healthcare network proposed by the Ministry of Health as a model for organizing the SUS, in which healthcare is divided by a thematic network in Brazil’s health regions [10]. In addition to the emergency care network, there are networks of points of care devoted to mother and childcare, psychosocial care, healthcare for persons with disability, and chronic disease care [10].

One prominent feature of emergency care services is care for acute and extremely severe problems, which involves a dynamic, heterogeneous work process and overload of activities, shortage of resources, and measures that can jeopardize workers’ physical integrity and health and contribute to increasing exposure to occupational risks [11,12].

During the pandemic, the longstanding issues involving nursing work, which favored significant risks of infection among workers, remained latent [13]. These risks now need to be acknowledged, and nurses’ perceptions in regard to these risks need to be identified.

## 2. Background

Health professionals were the main workforce in the fight against COVID-19. Therefore, they represented the most affected at-risk group for SARS-CoV-2 infection.

When the literature was consulted to investigate how to deal with the health issues of health professionals over the two years of the pandemic, a proposal for a public health decision support model to mitigate the spread of infections in Australia and New Zealand was found [14].

In Canada, a study discussed how fear of contagion, subjective overload, and perception of job insecurity and loss of income could directly affect physical and mental health [15].

A Brazilian paper, which was carried out based on research from published reports on health professionals who worked in the COVID-19 pandemic, mentioned that health-related periods of leave of absence were linked to the overload of work activities. Low testing in the initial period of the pandemic, due to a lack of the ideal number of tests, led to the occurrence of presenteeism and greater dissemination of the virus among this population [16].

After a review of the literature, it is evident that there is a lack of studies that include qualitative approaches with health professionals, especially nursing workers, in emergency services.

Even after several months of this scenario, a study carried out in a hospital in Southern Brazil identified a 30% increase in the absenteeism rate among these workers in this scenario, when compared to a follow-up study involving the last 25 years [17].

Previous studies do not address qualitative issues in the perception of workers. Thus, the following research question was formulated: Which strengthening and stressful factors influenced nursing workers’ health repercussions in emergency services during the COVID-19 pandemic?

Despite the volume of publications on issues relating to illness among health personnel during the COVID-19 pandemic and their related practical experiences, it was found that it was necessary to conduct this study in order to examine the health repercussions perceived by nursing workers working in emergency services during the onset of the COVID-19 pandemic in Brazil.

## 3. Materials and Methods

This qualitative, descriptive, cross-sectional study was based on interviews conducted from April to June 2020 at the onset of the COVID-19 pandemic in Brazil, in the setting of emergency services in Brazil. The research protocol was approved by the Brazilian National Council on Ethics on Research.

The studied population was nursing workers in emergency care services in Brazil, who were invited to take part in an interview and respond to a virtual sociographic questionnaire. Non-probability convenience sampling was used.

Health workers from all over Brazil were invited to participate in the survey; these were health workers who identified themselves as active in the face of suspected or confirmed cases of COVID-19. Those who did not complete the form or the answers to previously structured questions could be excluded.

Given the health restrictions in the workplace, the invitation to take part in the study was by sent out via email and social media networks. The snowball technique was used to reach participants. Members of the research group suggested the contact of the first participants, who indicated the next other informants, and so on, successively, until data saturation was achieved in the interviews.

Theoretical saturation was considered to be the degree of categories already developed, by seeking, through theoretical sampling, to saturate the identified codes [18].

Interviews were scheduled in advance at dates and times to suit informant availability. After the declaration of free and informed consent had been read and accepted by the participant, the interviews were conducted via videoconference and recorded, with the participant’s authorization, by using Google Meet^®^ and WhatsApp^®^ instant messaging or voice call.

Participants were invited to answer the main prompt: “Faced with this pandemic situation, tell me what it has been like to act as a health professional in your work environment”. If the objective of the study was not addressed in the first answer, the interviewer could use the auxiliary questions: “What can you perceive as factors that happen in your work process that strengthen the team?” and “What can you see as factors that happen in your work process that make it stressful for the team?”

Sociodemographic characteristics were collected using a form on the nine variables, which was emailed to each participant to be answered using the Google Forms^®^ platform. The variables were identified by number in the sequence of data collection, including the following variables: age, local region, job category, nature of the institution working with COVID-19, employment relationship with institution working with COVID-19, number of institutions at which participant worked, whether tested for COVID-19, health problems with prior medical diagnosis, and health problems mentioned. Participants who failed to complete the questionnaire were excluded. The sociodemographic variables were tabulated using Microsoft^®^ Excel^®^ for Microsoft 365 MSO (version 2301 Build 16.0.16026.20126) 64 bits and correlated to the respective interview.

From 96 realized interviews, 18 were from emergency service nursing workers, which is considered an adequate sample size considering data theoretical saturation [18].

The interviews were transcribed and checked by two independent researchers, identified by the word SUJ (subject), and numbered sequentially. It was decided to identify interviews by the word SUJ for analytical purposes, with no comparison between categories of participants, but rather seeking to understand their discussions. Job category information was applied only in the sociodemographic categorization used to profile the interviewees.

The interviews were then analyzed in sequence, using content analysis informed by Classic Lexicographical Analysis, Descending Hierarchical Analysis, and Similarity Analysis features of IRAMUTEC^®^ software, version 7.2 [18,19].

The series of interviews constituted a text corpus, which was treated so as to render its content comprehensible to the software. The text was saved in UTF-8 (Unicode Transformation Format 8-bit code units) character coding, which is the language recognized by the software [20].

The contents of the recording files remain stored for five years, as determined by the Brazilian research ethics legislation, in an external data storage device. Files were not stored in virtual data clouds in order to avoid any risk of intrusion and access to data.

## 4. Results

The study participants included 18 nursing workers, of whom 16 were nurses, one was a nursing auxiliary, and one was a nursing technician. Participants were profiled, as shown in Table 1.

Classic Lexicographical Analysis resulted in 18 texts, meaning that the 18 interviews were read by the software. It also found 12,068 words, 1338 active and supplementary forms, 625 hapax (words found only once in the corpus and without significant value), and a mean of 670.44 tokens per text.

First, words expressing positive and/or negative aspects of working in the COVID-19 pandemic were identified.

Prominent among the active forms from the text corpus were the words “strengthening” and “stressful”, with absolute frequencies of 50 and 19, respectively, expressing the positive and negative repercussions perceived by the participants.

The word “strengthening” revealed the positive aspects perceived by the workers. The discussions indicated that team unity, mutual support, reliable guidance, and service organization were essential to tackling the pandemic in the early months. The transcriptions below express these factors:


*[...] one strengthening factor was team union, which was already a strong point and was not daunted by the situation, because we were united [...].*

*(Subj 14)*



*[...] strengthening in the sense that the job is difficult and demand is heavy and the nursing team has been very united in its work [...].*

*(Subj 03)*



*[...] guidance was strengthening; it’s the question of whoever is doing the management of the place where I work [...].*

*(Subj 12)*



*[...] in my view, nursing management, nursing coordination showed strengthening. In the emergency sector where I work and in the hospital, we could see that they have done really useful work [...].*

*(Subj 16)*


The image produced by the similarity analysis, illustrated in Figure 1, shows how the word “strengthening” connected with the positive points encountered in the discussions.

The aspects identified as “stressful” stemmed from multiple factors relating to both the physical field and the mental and psychosocial field. The negative aspects perceived by the participants were as follows: lack of personal protective equipment (PPE), overwork, the work process, fear, patient suffering, and the stigma connected with the disease. The selected extracts portray these perceptions among nursing workers:


*[...] a stressful factor in the work process was the lack of PPE, which is something we are fighting for [...].*

*(Subj 07)*



*[...] the whole time there are thousands of pieces of information arriving on what can be used and what can’t be used, thousands of Ministry of Health norms, and we get left not knowing which way to go. That also causes very great stress [...].*

*(Subj 04)*



*[...] stressful because it’s a situation that causes an increase in overwork, because personnel end up missing work with suspected infection [...].*

*(Subj 06)*



*[...] stress is me being afraid to arrive home and pass it on to someone else, of being contaminated and asymptomatic. We take every care most of the time [...].*

*(Subj 01)*



*[...] I’m extremely stressed, very tired, physically and mentally, because we’re afraid [...].*

*(Subj 03)*



*[...] as wear and tear, there is this issue of the resistance of the team that is not part of the service.*

*(Subj 06)*


Figure 2 shows the connections between negative points that reflect these workers’ perceptions of stress.

The aspects identified as stressful in tackling the COVID-19 pandemic had repercussions on the health of the nursing workers operating in emergency care services. These repercussions were summarized in their discussions through the workers’ perceptions and could be identified by way of the text classification by their respective wordings by using Descending Hierarchical Classification (DHC).

By inserting the text corpus “Health repercussions perceived by nursing workers operating in emergency services and resulting from work during the COVID-19 pandemic”, the DHC returned a formulation of six classes which, after analysis, rendered it possible to form thematic areas, from which the three study categories emerged.

Figure 3 shows a dendrogram representing the formation of classes and, subsequently, the thematic areas. Note that, for purposes of significance, words with percentage frequencies > 45.00 and chi-square > 15.00 were used in the formation of the thematic areas.

The first category formed from class six was denominated “Emergency care service nursing workers exposure to risk of being contaminated with, falling ill from and transmitting the COVID-19 virus, as a result of lack of or improper PPE”. The discussions shown below expressed the emergency care service nursing workers’ perceptions consonant with the word “N95_mask”, which scored the highest in significance among the words listed in that class by the DHC:


*[...] these resources are not being made available to be used freely. Distribution is limited to equipment such as N95 masks, for instance, which are for individual use [...].*

*(Subj 16)*



*[...] we are far from the ideal, because PPE is restricted in the private system. I use the same N95 mask for 15 days. In the public system, I use the same N95 for 30 days and sometimes longer [...].*

*(Subj 02)*


The connection with words in classes five, two, and three brought out the second research category, which was expressed as “Changes in the environment, in work processes and relations in emergency care services resulting from tackling the COVID-19 pandemic”. This awareness was evidenced in the following passages:


*[...] as it is an emergency sector, the team is very small, not that multi-professional team they have in other sectors of the hospital. We managed to stay united as far as possible and help each other get through this phase [...].*

*(Subj 05)*



*[...] we were not ready. Even today, after three or four months, the flow is different every day, but we are never informed of what is going on. We dance to the music, that’s the reality of it [...].*

*(Subj 08)*



*[...] since the start of the pandemic, some structural changes have been made in my workplace and in relation to supply of PPE too [...].*

*(Subj 07)*



*[...] it is a constant challenge, many changes, questions of protocols [...].*

*(Subj 17)*



*[...] as it’s an unknown disease, you discover something different every day [...].*

*(Subj 10)*


Integrated analysis of classes four and one led to the formation of the last study category, termed *“Physical, mental and psychosocial alterations perceived by emergency service nursing workers in tackling the COVID-19 pandemic”.* The selected passages below establish the notion of wear in the three dimensions of changes in health:


*[...] from stress, I think the team put on at least half a kilo each. We ended up taking it out on food. What can you do? I’ve always been anxious, I have this emotional hunger. I’m eating a lot and sleeping a lot too [...].*

*(Subj 07)*



*[...] in addition to the exhaustion from having to work overtime because there’s no-one to put in place of whoever’s missing. So I think it’s tiredness and really quality of life. Smokers are smoking more, eaters are eating more [...].*

*(Subj 07)*



*[...] that overload has revealed itself as insomnia, anxiety, things that—as we spend a lot of time close together—things you didn’t see among our colleagues. I think that mainly insomnia, anxiety, insecurity have come out into the open a lot [...].*

*(Subj 05)*



*[...] mental health has been harmed, because people are at a very high level of stress, arguing a lot. We are a lot more intolerant. We’re not accepting a lot of things that are being proposed and so they end up being discouraging [...].*

*(Subj 08)*



*[...] staff who aren’t working on COVID don’t want to help. They’re afraid of touching patients. They treat patients and those who are*
*in the COVID unit as if they themselves were COVID [...].*

*(Subj 06)*


## 5. Discussion

A sociodemographic profile presents the same characteristics of nursing workers at the national level; it is remarkable that Brazilian nursing teams are composed of mid-level workers (technicians and nursing assistants) and higher-level workers (nurses). Although the teams are composed mostly of nursing technicians, in the interviews carried out in the study, the participation of nurses was predominant (88.88%) when compared to that of nursing technicians, which can be explained by the increase in the number of vacancies for nurses due to the complexity profile of COVID-19 referral centers [21].

According to the official data, the nursing workforce distributed in the 27 federative units of Brazil consisted of 2,565,116 professionals registered in the regional councils. Of these, 630,497 (24.58%) were nurses; 1,495,139 (58.29%) were nursing technicians; and 439,146 (17.12%) were nursing assistants [21].

A Brazilian study in 2016 showed that 77% of nursing workers were registered as nursing technicians or assistants, and only 23% were registered in the category of nurses. In 2021, data from the Federal Council recorded that nursing technicians and assistants totaled 75.40%, and nurses totaled 24.60%, which showed growth in the category of nurses in the previous five years [21,22].

The Brazilian nursing workforce is predominantly young, with an average age of 35 years. The study that presented the profile of nursing in the country pointed to the full rejuvenation of the profession, reporting 40% of its contingent to be aged between 36 and 50 years; (38%) to be between the ages of 26 and 35 years old, and 2% to be over 61 years old. Despite the proposition of a national study, the results indicated a greater participation of professionals registered in the southeastern region of the country [21,22].

With regard to emergency-care-service nursing workers’ exposure to risk of COVID-19 virus-related contamination, illness, and transmission resulting from a lack or insufficiency of PPE, inadequate supply of such equipment was the main source of stress among nursing workers, although supply is guaranteed by Brazilian Regulatory Standard #6 and constitutes one of the components of workers’ protection. An alert issued by the Ministry of Health pointed to the difficulties inherent to a lack of supplies and limited human and material resources [12,13,23]. The scarcity of PPE exposed health workers to high levels of stress and required them to make serious decisions, as did the limited supply of ventilators for maintaining care for patients affected by the pandemic [13].

Emergency service nursing personnel were observed to report that they were exposed because of scarce, unsuitable, or incomplete PPE, which caused discomfort and difficulty in providing care to patients and reinforced their fear of contamination [1,7,24].

Added to the issues involving supply of PPE, uncertainty about the virus (transmission routes, incubation time, lethality, reliable diagnostic methods, and effective treatment) also affected the health of nursing personnel [1,25].

In China, which was the first country to report cases of the disease, contamination of workers at the onset of the pandemic was favored by improper protection, which can be explained by a lack of knowledge about the pathogen and frequent and prolonged exposure to potentially contaminated patients [4].

In addition to facing the fear of falling ill and of transmitting the disease, uncertainty about the care being provided, a lack of supplies, a lack of PPE, and management difficulties, nursing workers had to deal with issues involving attitudes toward prevention measures in society at large, all of which culminated in changes in how routine work activities were carried out [2,26,27].

The endeavor to combat the COVID-19 pandemic resulted in changes in emergency care service work environments, processes, and relations. The necessary arrangements and changes in physical installations at points of care for suspected or confirmed cases of COVID-19 formed part of a strategy to prevent and control infection. However, these alterations were responsible for causing physical and mental fatigue because they were introduced intensively as new recommendations emerged [2,27].

Nurses reported perceiving an impact on their job functions, including increased demands, and reported performing tasks outside of their everyday functions, all of which were combined with a lack of respite and rest during shifts [28].

Deficiencies in physical structures and material resources, a lack of suitable places to rest and eat, disorganization, and unsuitable working environments were situations experienced by these workers, which—over and beyond exposure to COVID-19—constituted situations of institutional violence to which nurses were exposed. Even though these conditions were not perceived explicitly by the study participants, they can be identified in declarations regarding unfavorable work environments in the interviews [13,29,30].

A Brazilian study which included various health professional categories examined working conditions of frontline personnel. One preliminary finding was that 43.2% of health workers felt unprotected in their work in combating the COVID-19 pandemic [31].

In July 2020, a set of guidelines was published for a matrix of care pathways in various COVID-19 care scenarios. Regarding emergency services, the guidelines recommended reviewing work processes and organization and proposed expanding and/or modifying care areas and establishing flows, policies, and procedures, in addition to forecasting strategic inputs [32].

The recommendations go beyond the work process to focus on measures to protect personnel, such as planning workload to guarantee appropriate working hours, rest periods and mandatory intervals, proper rest areas, and workflows that maintain physical distancing between workers, so as to prevent physical and mental exhaustion and errors, as well as fostering a decent working environment [33].

In the past two years, discussions of the nursing work process and its organization have not been limited to COVID-19 because issues of mental suffering, dehumanization, and naturalization of suffering at work all worsened in the period [3].

The physical, mental, and psychosocial alterations perceived by emergency service nursing workers while responding to the COVID-19 pandemic have been identified in WHO publications, indicating that nursing workers worldwide were pressured by the pandemic situation, which caused severe health problems, particularly mental health problems [34].

These changes have also been described in the experiences of nursing personnel while responding to the pandemic in China, Turkey, Italy, Canada, United States, Philippines, Brazil, Portugal, and Ethiopia. The descriptions indicated that the main health effects were anxiety, depression, stress, post-traumatic stress disorder, mental sleep disorders, and burnout [35].

The risk of COVID-19 infection impacted the mental health of nursing workers especially, which undermined the attention and decision-making ability of these workers, who expressed reactions common to the public at large and specific to their professional experience [26,34].

Corroborating the aforementioned finding, the virtual care offered by the COFEN found that nursing workers from various areas in Brazil expressed their feelings with regard to the issues raised by the pandemic, revealing physical, mental, and psychosocial alterations that led to illness [24].

These emotional responses are understood to be associated with individual and collective coping mechanisms and influence both the way people in general act and, more particularly, the actions of health personnel [35].

Although mental alterations have been an area of concern, Brazilian and international studies also note prominent physical alterations, among which are cardiovascular conditions; nervous system-related smell and taste dysfunctions; neuromuscular dysfunction; conditions such as Guillain–Barré syndrome; clinical gastrointestinal effects such as diarrhea, anorexia, vomiting, nausea, abdominal pain, and complications such as gastrointestinal bleeding; and musculoskeletal sequelae involving weakness resulting from loss of muscle strength, immobility, and malnourishment [36,37].

The COVID-19 pandemic revealed the public’s ambivalence toward health personnel, especially frontline workers combating the virus. Although on the one hand, health personnel were applauded, there were also episodes of repulsion and discrimination against nursing professionals, in scenarios in which contact with them was shunned, which favored the psychosocial alterations developed by these workers [24].

The changes found in this study are in agreement with those most commonly described elsewhere during pandemics: lack of or excessive appetite, insomnia, sleeplessness or over-sleeping, recurrent nightmares, interpersonal conflicts with members of family or work teams, violence against health personnel, and recurrent thoughts about the pandemic, their family’s health, and death and dying [26].

Work overload and the number of informal contracts in this pandemic context led to significant changes in Brazilian workers’ daily lives, including sleep disturbance, frequent irritability, inability to relax, stress, loss of career or life satisfaction, sadness, and apathy [31].

It has become fundamentally important to think carefully about introducing effective measures to promote and protect nursing workers’ health in light of the challenges experienced in combating COVID-19 and the diversity of situations posed in Brazil [3,32]. It has also become essential to examine the perceptions of these workers, especially those working directly on the frontline in crisis scenarios in which the risk of infection is real and should be managed in such a way as to protect the workers involved [2,5].

After this study was carried out, the persistence of old problems in nursing and health work was observed, ranging from the purchase of protective materials, personnel dimensioning, respect for health professionals’ restrictions in dangerous environments, and even to the need of psychosocial support.

Nursing workers must dedicate themselves to work supported by robust scientific evidence, not only for the promotion of healthcare but also to demand the necessary structure to carry out their activities, guaranteeing the promotion of health and safety in the work environment.

In addition, training during professional education and after entering a daily work routine promotes the development of knowledge, skills, and a positive attitude to respond to public health emergencies, in addition to promoting greater involvement and job satisfaction, increasing rates of satisfaction and the improvement of the capacity of management [2,38,39,40].

## 6. Conclusions

This study reported the perceptions of nursing workers in emergency services in Brazil, which pointed to exposure-related issues (the risk of their being contaminated, falling ill, and transmitting the COVID-19 virus) resulting from a lack of or inappropriate PPE, in addition to alterations in work environments, processes, and relations in emergency care services, which had physical, mental, and psychosocial repercussions on these workers’ health.

The physical alterations that were identified were disturbances in dietary patterns, physical tiredness, fatigue, and heavier smoking. The mental alterations reported in the interviews were anxiety, sleep pattern alterations, fear, exhaustion, and stress. The psychosocial alterations cited by participants were social isolation, loneliness, distancing from families, and social stigma from working in a sector devoted to COVID-19 care.

With regard to limitations, there was a selection bias, since only individuals with better access to technological tools and internet connectivity could participate in the survey to answer the sociodemographic form and the interview electronically. In addition, it was found that despite the research having a national scope, only workers from the North and Southeast regions participated. We suggest further research in workplaces as pandemic control advances.

This study presents contributions with regard to workers’ health, since it identified their physical, mental, and psychosocial illness through the repercussions listed by the workers themselves from their experience in service. This study also presented data on the illness of nursing workers who worked in emergency services at the beginning of the pandemic in Brazil.

The pandemic re-signified the public’s regard (and that of the nursing workers themselves) for the importance of their profession, the risk inherent to their work, and the repercussions on these workers’ health.

## Figures and Tables

**Figure 1 ijerph-20-04675-f001:**
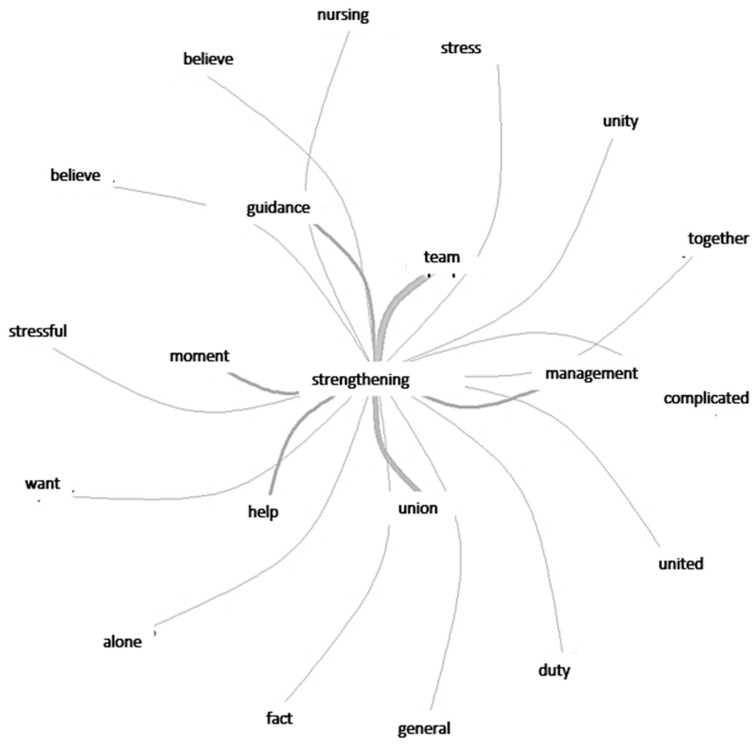
Graphic representation of words expressing strengthening factors.

**Figure 2 ijerph-20-04675-f002:**
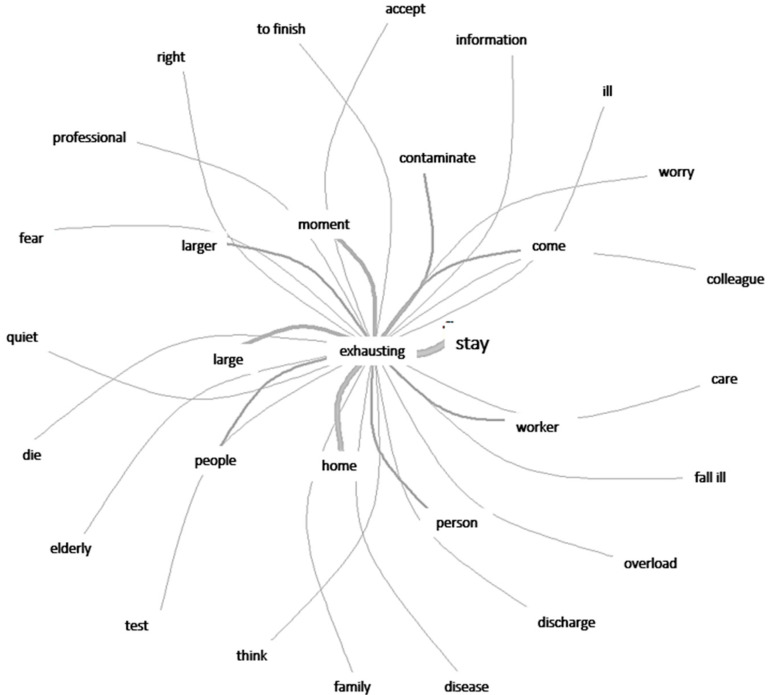
Graphic representation of words that express stress factors.

**Figure 3 ijerph-20-04675-f003:**
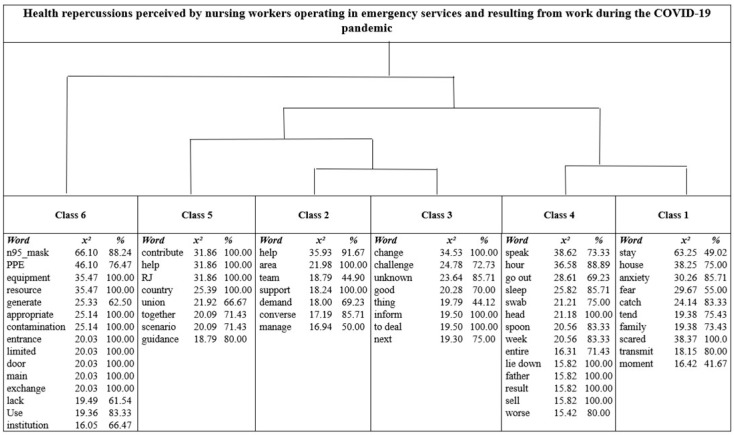
Dendrogram showing words that formed the study categories. Note: RJ = Rio de Janeiro.

**Table 1 ijerph-20-04675-t001:** Study participants’ sociodemographic, employment, and clinical characteristics, Rio de Janeiro, Brazil, 2020 (*n* = 18).

Variables		*n*	f (%)
Age (years)	20–29	7	38.89
Minimum: 25	30–39	7	38.89
Maximum: 52	40–49	3	16.67
Mean: 34.77	50–59	1	5.55
Local region	Southeast	15	83.33
	North	3	16.67
Job category	Nurse	16	88.88
	Nursing auxiliary	1	5.56
	Nursing technician	1	5.56
Nature of the institution	Public	15	83.33
	Public/Philanthropic	1	5.55
	Public/Private	2	11.12
Employment relationship	Public sector statutory	7	38.89
	Temporary contract	5	27.78
	Temporary contract/Private contract	3	16.67
	Public sector statutory/Private contract	2	11.11
	Private contract	1	5.55
Number of institutions at which work	1	6	33.33
	2	11	61.11
	3	1	5.56
Tested for COVID-19	No	9	50.00
	Yes	8	44.44
	No information	1	5.56
Prior health problems	Yes	9	50.00
	No	5	27.78
	No information	4	22.22
Problems reported	Systemic arterial hypertension	3	16.66
	Diabetes mellitus	1	5.55
	Obesity	1	5.55
	Herniated disk	1	5.55
	Anxiety	1	5.55
	Bronchitis	1	5.55
	Epilepsy	1	5.55
	Insulin resistance	1	5.55
	Prolactinoma	1	5.55

Font: Author’s data.

## Data Availability

The data are not publicly available. A deidentified dataset can be obtained by contacting the first author.

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
