# Peer review of "COVID-19 Pandemic: Analysis of Health Effects on Emergency Service Nursing Workers via a Qualitative Approach"

_ijerph, 2023, doi:10.3390/ijerph20064675_

Round 1

Reviewer 1 Report

Optional. Include in the title the research design.

Consider not including researchers in the sample

Author Response

Dear reviewer,

We appreciate your conisderations, which contributed to the improvement of the manuscript. Below, we present the responses to your notes.

English language and style

(x) English language and style are fine/minor spell check required: We declare that the text was translated by a native English speaker

Is the research design appropriate? – Consider not including researchers in the sample - review was performed to improve understanding. Researcher were not in the sample, but gave first contacts to be interviewed.

Optional. Include in the title the research design – included as suggested.

We remain at your disposal for further clarifications or revisions, if necessary.

Best Regards.

Reviewer 2 Report

The article is very informative, interesting, and easy to read. The result section is well done. 

Questions and Suggestions:

A. Under 'Materials and Methods'

1) Regarding the recruitment of participants using the snowball technique (lines 99-103), how did you determine that 'saturation was achieved? Was your decision based on theoretical or data saturation? I believe that some of your readers may want to know this. 

2) The paper did not specify the nature of the interview questions. Were these structured or open-ended? I don't think that I read this under Materials and Methods.

3) Knowing that the interviews were recorded, for confidentiality purposes, how did you handle the possibility of matching the identity of participants with the interview recordings? What happened to the recordings post-transcription? 

B. Under "Results"

 On page 7,  Figure 2, Class 5, what does RJ mean? Consider using something readers will understand or provide key to explain what the abbreviation stands for.

C. Under 'Discussion'

Page 8, paragraph 2, consider adding % to the workforce distribution of the 27 federative units. Makes for consistency.  

D. General Errors. 

These are related to sentence structure as well as typographic errors. 

1) Abstract. Review the very first sentence for completeness. 

2) Long sentences. Found several long sentences with resulting loss of meaning. These could also lead to loss of reader's interest.  Consider making sentences shorter. Refer to "Materials  and Methods", lines 104 to 107, and 108 to 116. Consider also re-structuring lines 344 to 350 and 372 to 377.  Do look through the article for others.    

3) Inconsistencies. 'Nursing staffs' and 'nursing workers' are used interchangeably. Stick with one.

4) Line 101, 'The snowball technique... candidates participants'. You can delete 'candidates'.

Author Response

Dear reviewer,

We appreciate your considerations, which contributed to the improvement of the manuscript. Below, we present the responses to your notes.

English language and style - (x) Moderate English changes required: We declare that the text was translated by a native English speaker

A. In 'Materials and Methods'

1) Regarding the recruitment of participants using the snowball technique (lines 99-103), how did you determine that 'saturation was achieved? Was your decision based on theoretical or data saturation? I believe that some of your readers may want to know this. – our decision was based on theoretical saturation. This information was included in the manuscript (lines 141-2).

2) The paper did not specify the nature of the interview questions. Were these structured or open-ended? I don't think that I read this under Materials and Methods. – structured questions were used. We insert this information between lines 148-53.

3) Knowing that the interviews were recorded, for confidentiality purposes, how did you deal with the possibility of confronting the identity of the participants with the recordings of the interviews? What happened to the recordings after transcription? – characterization form was completed in a separate form, with the interview identified only by the sequence number (line 120). The contents of the recording files remain stored for five years, as determined by the Brazilian research ethics legislation, in an external data storage device. Files were not stored in virtual data clouds in order to avoid any risk of intrusion and access to data (lines 177-9).

B. Under "Results"

On page 7,  Figure 2, Class 5, what does RJ mean? Consider using something readers will understand or provide key to explain what the abbreviation stands for. – It means “”Rio de Janeiro”. However, many people speaks that way. We included a note under explaining that (line 250).

C. Under 'Discussion'

Page 8, paragraph 2, consider adding % to the workforce distribution of the 27 federative units. Makes for consistency.  – it was included

D. General Errors.

These are related to sentence structure as well as typographic errors.

1) Abstract. Review the very first sentence for completeness. – it was reviewed

2) Long sentences. Found several long sentences with resulting loss of meaning. These could also lead to loss of reader's interest.  Consider making sentences shorter. Refer to "Materials  and Methods", lines 104 to 107, and 108 to 116. Consider also re-structuring lines 344 to 350 and 372 to 377.  Do look through the article for others. - sentences were revised by the translator

3) Inconsistencies. 'Nursing staffs' and 'nursing workers' are used interchangeably. Stick with one. – we consider the expression "workers" more appropriate to the field of knowledge.

4) Line 101, 'The snowball technique... candidates participants'. You can delete 'candidates'. – it was reviewed.

We remain at your disposal for further clarifications or revisions, if necessary.

Best Regards.

Reviewer 3 Report

This an important and interesting article. Especially since there is still a lack of studies on this topic, however there a some major concerns about this paper in its current form.

There is no background section - from Introduction, the authors skip to materials and methods. Hence, the authors need to add such a section right after the introduction, where they will explain what past research has said regarding them own investigation

Based on this new section, the authors need to then state their research question more clearly and/or formulate their hypothesi(e)s accordingly. Some of this information has already been included in the materials and methods section, and in the discussion. However, they need to be utilized in section 2 (background), in order to justify the importance and focus of this research clearly (before explaining how the research was actually made). In essence, the background section needs to be added, to explain what existing insight in the literature has led lead to the investigation of what research question and/or specific hypothesi(e)s.

The article is strongly original in the subject, and appreciable in the theoretical contextualization. But its main problem arises from the methods too weak for a this kind of research

I imagine this work as a sort of appreciable pilot study bound to test questions and hypothesis. But the true research should be then based on a rather different sampling strategy. 

Has an assessment of the quality of the studies included been made? If yes, please provide data on the scales and the cut-off used. This part should be addressed in the results section also.

Please provide more information on the inclusion and exclusion criteria used to assess the eligibility and describe more the process of methodology. In this form is very week. You have a big team of authors so try to prepare this paper in the proper way.

The theoretical and practical contribution must be expanded. More specifically, the authors can add some more specific insight on how their theoretical and practical insight can be utilized by future researchers and practitioners. E.g. they state that "This article revealed the gaps and the need for training and education to improve future nurse preparation in crises/emergency.". Some more information can be added and/or suggestions on how these findings can be put to practice in the future.

The limitations have not been adequately explored. Moreover, no suggestions for future work have been made according to the outlined limitations. 

Also, there is no theoretical framing of the argument. The authors need to engage with the literature on what affects nurse, before they can made any deductions on whether this has improved or not during and after COVID and why it is important to understand this.

Please see and include following references:

https://doi.org/10.3390/ijerph18031348

https://doi.org/10.1371/journal.pone.0244488

https://doi.org/10.1371/journal.pone.0277484

https://doi.org/10.3390/ijerph181910545

https://doi.org/10.1016/j.nepr.2022.103327

Also I advise to add more literature by your own.

 I strongly recommend careful revision by an experienced editor in English and do not use Spanish in your manuscript. 

I’m really looking forward for your revised work as I feel this paper has merit.

Author Response

Dear reviewer,

We appreciate your considerations, which contributed to the improvement of the manuscript. Below, we present the responses to your notes.

There is no background section - from Introduction, the authors skip to materials and methods. Hence, the authors need to add such a section right after the introduction, where they will explain what past research has said regarding them own investigation – it was reviewed and included. (lines 93-113).

Based on this new section, the authors need to then state their research question more clearly and/or formulate their hypothesi(e)s accordingly. Some of this information has already been included in the materials and methods section, and in the discussion. However, they need to be utilized in section 2 (background), in order to justify the importance and focus of this research clearly (before explaining how the research was actually made). In essence, the background section needs to be added, to explain what existing insight in the literature has led lead to the investigation of what research question and/or specific hypothesi(e)s. – it was reviewed and included (lines 114-121).

The article is strongly original in the subject, and appreciable in the theoretical contextualization. But its main problem arises from the methods too weak for a this kind of research

I imagine this work as a sort of appreciable pilot study bound to test questions and hypothesis. But the true research should be then based on a rather different sampling strategy. - we understand the weaknesses of snowball sampling, but at the height of the spread of SARS-CoV-2 this was the only possible way to approach health professionals, given the constraints of isolation and social distancing. Even so, it is able to produce evidence and be discussed and even deepened in new scientific investigations

Has an assessment of the quality of the studies included been made? If yes, please provide data on the scales and the cut-off used. This part should be addressed in the results section also. – structured questions were used, not scales. This information was inserted in the method session (lines 148-153).

Please provide more information on the inclusion and exclusion criteria used to assess the eligibility and describe more the process of methodology. In this form is very week. You have a big team of authors so try to prepare this paper in the proper way. – Reviewed and included in lines (131-134,163).

The theoretical and practical contribution must be expanded. More specifically, the authors can add some more specific insight on how their theoretical and practical insight can be utilized by future researchers and practitioners. E.g. they state that "This article revealed the gaps and the need for training and education to improve future nurse preparation in crises/emergency.". Some more information can be added and/or suggestions on how these findings can be put to practice in the future. Included – lines 426-438

The limitations have not been adequately explored. Moreover, no suggestions for future work have been made according to the outlined limitations. – suggestion was included (lines 455-6).

I strongly recommend careful revision by an experienced editor in English and do not use Spanish in your manuscript - We declare that the text was translated by a native English speaker.

About suggested references:

https://doi.org/10.3390/ijerph18031348 - we don’t consider that this reference if appropriate to the study, once the discussion of a health measurement tool is not in the object of this qualitative study.

https://doi.org/10.1371/journal.pone.0244488 - we don’t consider that this reference if appropriate, once “disasters” contemplates a high variety of events and not necessarily epidemics/pandemics.       

We remain at your disposal for further clarifications or revisions, if necessary.

Best Regards,

Round 2

Reviewer 3 Report

Thanks for including all my comments. I think paper has been improved and might be published however I still think Spanish names in text are not needed

Author Response

Dear reviewer,

We thank you for your considerations and inform you that we have withdrawn the descriptions in Portuguese of Brazilian policies.
New changes are highlighted in yellow.

Best Regards.